# Modelling the Interaction between a Laterally Deflected Car Tyre and a Road Surface

Algirdas Maknickas [1,*,†], Oleg Ardatov [1,†], Marijonas Bogdevičius [2,†] and Rimantas Kačianauskas [1,3,†]

1   Faculty of Mechanics, Vilnius Gediminas Technical University, LT-10223 Vilnius, Lithuania
2   Faculty of Transport Engineering, Vilnius Gediminas Technical University, LT-10105 Vilnius, Lithuania
3   Faculty of Civil Engineering, Vilnius Gediminas Technical University, LT-10223 Vilnius, Lithuania
*   Correspondence: algirdas.maknickas@vilniustech.lt
†   These authors contributed equally to this work.

**Abstract:** The interaction between a deflected car tyre and a road surface was modelled under normal and overloaded conditions. The model incorporates the detailed geometry of the car wheel, including a metallic rim reinforcement, a hyper-elastic composite tyre and tyre treads, as well as the geometry of the road surface. The finite element method was used to investigate the nonlinear dynamics of the model and study the influence of tyre deflection on the friction coefficient of the tyre tread. The results were compared under different internal tyre pressures and external wheel loads, including underloaded, normal and overloaded states. The skewness of the distribution of friction coefficients was calculated under these different conditions. The results show that overloading and nonoptimal internal tyre pressure cause reduced and skewed friction, with the asymmetry in the tyre–road contact zone resulting in a moving instability of the car during braking.

**Keywords:** deflection; FEM; friction coefficient; modelling

## 1. Introduction

Car safety is influenced by many mechanical parameters, the most important one being the friction force between the wheel and the driving surface. This friction force depends on three general factors: the multi-scale roughness of the tyre's contact zone, the forces acting in the contact zone with the road surface and the coefficient of friction between the tyre and the road surface. Additional important factors include the tyre deflection area and the distribution of forces in the wheel–road contact zone.

Maintaining car safety with increasing intensity of city traffic requires the consideration of many factors, including some secondary effects that may seem insignificant at first glance. It is clear that tyre–road friction plays a key role in vehicle handling performance. Understanding the behaviour of a vehicle in frictional motion requires the establishment of the relationship between various parameters, such as the pressure distribution and the shape and size of the contact patch.

A comprehensive overview of the state of the art in the field of vehicle and tyre state estimation is presented in the review paper of Singh et al. [1]. This paper, limited to automobile wheels, presents information about different techniques proposed in the literature for estimating the tyre forces, the vehicle lateral velocity and the sideslip angle. A detailed description of the interaction factors and parameters may be found in [2], which discusses the evaluation of forces in the wheel–road contact zone. In addition, the importance of deformed tyre shape to driving safety is demonstrated.

The tyre–road interaction is very complex, and many variations of this phenomenon, with different kinds of tyres and road surface structures and properties, have been investigated in numerous theoretical and experimental studies.

An analysis of contact patch sizes (i.e., length and width) and tyre deflection due to the resistance of soil at different wheel loads and inflation pressure levels was presented in [3].

An approach to predicting the slippage of a tractor's driving tyres, including a numerical expression of the tyre inflation pressure, was discussed in [4].

Jaime Hernandez [5] used a hyper-elastic model of a tyre to predict energy in the tyre and contact stresses. A study of friction in the case of rubber sliding on smooth glass, concrete and asphalt road surfaces was presented by Tolpekina et al. [6].

There have been significant efforts to understand tyre–soil interactions. Janulevičius et al. [7] studied the dependence on wheel pressure and external vertical loads in the case of wheel interaction with a terrain surface. Jonah H. Lee and Krystle Gard [8] presented stochastic soil–tyre models which could serve as tools for predicting model responses due to various uncertainties. The tyre–sand interaction has been investigated by several authors [7–10]. A tyre–sand interaction model was also proposed in [11] and showed good agreement with experimental data. Junya Yamakawa et al. [10] presented experimental data on the motion of a vehicle travelling on dry sand. Their work also contains comparisons with theoretical research; the values of tyre forces are also given.

Several papers investigated snow-influenced conditions [12–15]. Lee [12] calibrated and validated a tyre–snow interaction model, Choi et al. [13] presented a numerical investigation of snow traction characteristics, Walus et al. [14] presented a general analysis of tyre–road contact in winter conditions and Shoop et al. [15] performed modelling of tyres on snow. The work of Giessler et al. [16] was focused on the temperature dependence of traction on ice and snow. These authors also derived the limiting curve of force transmission on ice and snow. In [17], thermal effects and a strong correlation between thermal conditions and transmittable forces on a sand road were found by monitoring temperatures on a tyre surface using a high-speed infrared camera.

Most physically adjusted theoretical relationships are of semi-empirical character, that is, they were obtained based on experimental results. Due to the large number of parameters, experimental validation of theoretical results can be problematic. Generally, the most important tyre parameters are estimated by manufacturers by conducting tyre dynamics tests. However, these tests cannot fully characterise the tyre in special conditions, for example, dynamic behaviour under low inflation pressure [18]. Due to the high costs of experimental tests, numerical simulation has become an important alternative to physical testing.

Substantial progress in computer hardware and the improvement of numerical discrete analysis methods has stimulated the development of advanced software tools able to solve complicated physical problems, including tyre–road interactions. Novel software technology makes theoretical analysis methods more feasible. Physical data that have not been measured experimentally may be inferred using numerical analysis.

However, the numerical analysis of tyres involves several difficulties stemming from geometric complexity and the use of materials exhibiting a high degree of nonlinearity. The remarkable differences in the sizes of tyre structures cause the most trouble in the creation of numerical schemes. The required fine discretisation of the tread part of the tyre substructures and inhomogeneous road or pavement layers in the contact zone become more crucial in the 3D case.

Different approaches have been proposed for discretisation of both tyre and terrain and their coupling. The characteristics of tyres on wet surfaces were examined by El-Sayegh and El-Gindy [19]. Their work used the finite element method (FEM) and the smoothed-particle hydrodynamic method. The multiscale approach presents the most general strategy formulated by coupling the discrete element method and the FEM models on the contact surface. It enables simulation of the behaviour of particulate terrain, such as soil or ice, by discrete element methods, while using FEM for accurate modelling of a tyre as a continuous body [20,21]. Hyunggyu Jun et al. [21] proposed a coupled approach for the investigation of tyre–sand interaction, and Ge et al. [2] and Bui et al. [22] proposed a Lagrangian meshfree particles method for large deformation and failure flows of geomaterial with the development of an elastic–plastic soil constitutive model.

Many studies have established tyre models using FEM to predict tyre–road contact stress or to evaluate tyre performance. The truck radial tyre model developed in [23] and the 205/55R16 tyre model developed in [18] are examples. The main challenge generating 3D FEM meshes is the description of the complex pattern of a tyre tread composed of many grooves and blocks. An effective mesh generation procedure for 3D automobile tyres was proposed in [24].

When a vehicle is braking, the front wheels are subjected to more vertical force than the rear wheels. Therefore, different friction forces are applied to the front and rear wheels, which affects the braking efficiency of the vehicle.

A tyre can be classified by the curvature of the tyre contact zone geometry in three different shapes [25]: underdeflected, ordinary deflected and overdeflected tyre shapes; an underdeflected tyre shape occurs when the value of the tyre's internal pressure is abnormally high, an ordinary deflected tyre shape occurs with a correct internal tyre pressure, guaranteeing a maximal surface area for the wheel tyre–road contact zone, and an overdeflected tyre shape can occur due to insufficient tyre pressure or overloading by additional external force.

This study extends our previous work [26] and concentrates on modelling the distribution of the friction coefficient throughout the tyre tread. The overall aim is to determine the friction coefficient distribution in the tyre–road contact zone by calculating the normal, tangential longitudinal and lateral contact forces for the laterally deflected tyre with the road surface for different internal tyre pressures and external loads, focusing on the front wheels of a braking car. The article has four sections: Introduction, The tyre–road model, Results and discussion and Conclusions, where model section has in addition the subsections Problem description, Tyre–road geometry, Mechanical properties, Tyre material model, Contact interaction asymmetry and Tyre–road finite element method.

## 2. The Tyre–Road Model

### 2.1. Problem Description

The problem of dynamic contact between tyre and road in the case of sudden braking was considered. In order to solve it, the nonlinear theory of elasticity was applied. It was assumed that the tyre acts like a hyper-elastic Mooney–Rivlin material, while the road surface was modelled as a perfectly elastic continuum with zero roughness. To capture the realistic deflection of the tyre in the moment of braking, a 3D model of the wheel, including the tyre and rim, was developed. The dimensions and the asymmetrical tread of the tyre were taken from real examples that we investigated in previous work [9].

We set three different values, 0.18, 0.2 and 0.22 MPa, for the tyre's internal pressure. The road surface was shifted 30 mm from a contact zone in normal (negative y) direction of the fixed wheel. The tyre's external load was determined by modelling the compression force as a function of base displacement, when the wheel was rigid and the road surface moved in a direction normal to the wheel contact zone.

The model movement steps are presented graphically in Figure 1. The initial loading conditions were applied for 10 s. In the first second, pressure was applied to the tyre (depending on the study, from 0.18 MPa to 0.22 MPa). The pressure delivery graph is shown in Figure 1b. Then, a vertical displacement toward the inflated tyre was applied for the component representing the asphalt concrete surface. At the ninth second of the test, the vertical displacement was stopped and the asphalt component moved tangentially to the wheel direction in order to evaluate the slip of the tread in the contact zone and to determine the friction forces (see Figure 1a). This movement lasted 1 s and was applied in order to obtain the values of the tangential force. In addition, the wheel was continuously subjected to centrifugal forces (an angular velocity of 80 rad/s was assumed).

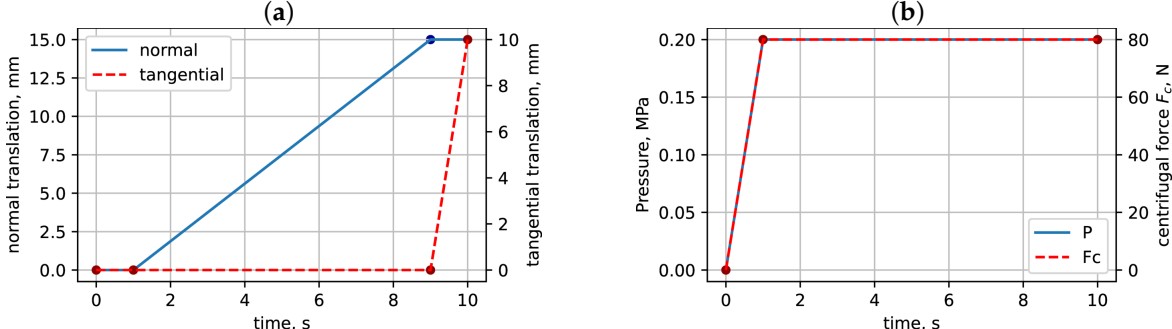

**Figure 1.** Model time steps: (**a**) movement of road surface and (**b**) pressure in tyre vs time.

Next, to clarify the observations, all tyre tread blocks that contacted the modelled road surface were labelled. The tyre tread labelling profile is given in Figure 2a, and the moving directions of the tyre and road interface are given in Figure 2b.

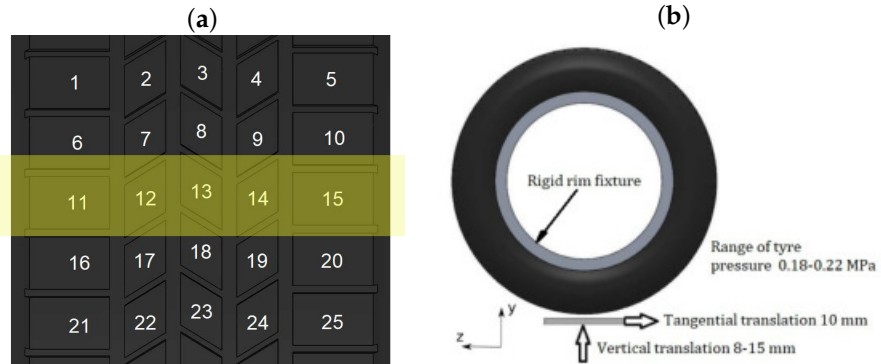

**Figure 2.** (**a**) Labelling of tyre tread blocks that contact the road surface and a selected centre line. (**b**) Movement directions of model parts.

To verify the mechanical behaviour of the tyre–road model, the nonlinear theory of elasticity was used and the following equilibrium equations of the dynamic system at the each time step were solved. The 3D nonlinear analysis was performed using an implicit time integration Newmark-beta scheme and a Newton iterative method in the SolidWorks software [27].

### 2.2. Tyre–Road Geometry

The geometry of the model is presented in Figure 3. It has four parts: the tyre, the rim, the elastic insert and the road surface. The model was developed from the real geometry of an R16 car tyre, including treads, using the Solidworks 2020 CAD software [27]. Simplification of the rim geometry was required to reduce the total number of finite elements. As shown in Figure 3, part 1 and part 3 denote two layers of the tyre, where the rubber is part 1 and the elastic insert is part 3. Finally, the road surface interacts with the tyre during the simulation (part 4 in Figure 4).

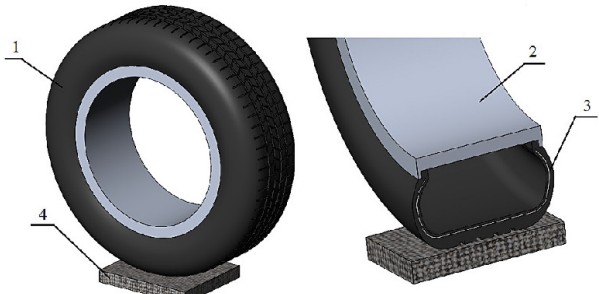

**Figure 3.** Geometry of the model: 1—tyre, 2—rim, 3—elastic insert and 4—road surface.

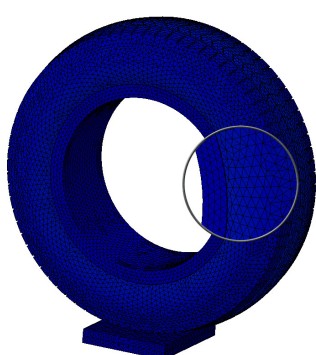

**Figure 4.** The tyre–road finite element mesh.

### 2.3. Mechanical Properties of Components

The tyre has a layered structure, in which rubber is the main material and a thin layer of metal is placed inside. The inside layer of the tyre–road system was modelled as an elastic material. The Mooney–Rivlin hyper-elastic material model was used to describe the rubber, with the values of the constants set at $C_{10} = 5.79$, $C_{01} = 1.45$, Poisson's ratio $\nu = 0.4995$ and the density $\rho = 1200 \, \text{kg/m}^3$. The rim and road surface were also modelled as elastic materials. The elastic constant values of the whole system are given in Table 1.

**Table 1.** Elastic constants of the tyre–road system.

| Part | Material | Elastic Modulus, MPa | Poisson Coefficient |
|---|---|---|---|
| Rim | Aluminium | 69,000 | 0.33 |
| Elastic sub-layer | Composite | 200 | 0.394 |
| Base | Road surface | 4800 | 0.28 |

All values of the material constants were set according to the results of physical tests with the experimental tyre until proper agreement between numerical and physical modelling was obtained.

### 2.4. Tyre Material Model

The strain energy density function of a Mooney–Rivlin material is expressed as a two-constant formulation [28],

$$W = C_{01}(I_1 - 3) + C_{10}(I_2 - 3) + \frac{1}{2}K(I_3 - 1)^2, \tag{1}$$

where $C_{01}$ and $C_{02}$ are the first and second material constants, respectively, related to the distortional response, and $K$ is the material constant related to the volumetric response. $I_1$, $I_2$ and $I_3$ are the reduced invariants of the right Cauchy–Green deformation tensor, which depends on the principal stretch ratios.

The material constant $K$ can be determined as

$$K = \frac{6(C_{01} + C_{10})}{3(1 - 2\nu)}, \tag{2}$$

where $\nu$ is Poisson's ratio.

### 2.5. Contact Interaction Asymmetry

The contact interaction asymmetry was investigated by applying two mechanical parameters: the Fisher–Pearson coefficient of skewness and the ratio between the left-side-labelled tread block subtotal sum of forces and the right-side-labelled tread block subtotal sum of forces in the contact zone.

The Fisher–Pearson coefficient of skewness, or sample skewness, is computed as

$$g_1 = \frac{m_3}{m_2^{\frac{3}{2}}} \tag{3}$$

where $m_i$ is the sample $i$-th central moment,

$$m_i = \frac{1}{N} \sum_{n=1}^{N} (x_n - \overline{x})^i, \tag{4}$$

where $\overline{x}$ is the sample mean.

The ratio of left-side-labelled tread block subtotal sum and right-side-labelled tread block subtotal sum of normal forces and the same ratio for tangential transverse forces were calculated using

$$AS_k = \frac{\sum_{i=1}^{N} F_{ki}}{\sum_{j=1}^{N} F_{kj}}, \tag{5}$$

where $i \in [1, 2, 6, 7, 11, 12, 16, 17, 21, 22]$ are the indices of the forces on each left-side tread of the tyre, $j \in [4, 5, 9, 10, 14, 15, 19, 20, 24, 25]$ are the indices of the forces on each right-side tread of the tyre and $F_k$ denotes force normal component $F_n$ and tangential transverse component $F_{tx}$ (see Figure 2).

### 2.6. Contact Friction Coefficient

The contact friction coefficient on the 2D road can be expressed as [29]

$$\mu = \frac{\sqrt{F_{tx}^2 + F_{tz}^2}}{F_n} = \sqrt{\mu_x^2 + \mu_z^2}, \tag{6}$$

where $F_n$, $F_{tx}$ and $F_{tz}$ are the normal, tangential transverse and tangential longitudinal force components, respectively; $\mu_x = F_{tx}/F_n$ and $\mu_z = F_{tz}/F_n$.

### 2.7. Tyre–Road Finite Element Mesh

A mesh of tyre, rim (Figure 4), elastic inlet and basis was created using the software SOLIDWORKS-2020 [30]. It was used to create a solid mesh with parabolic—also called 'second-order' or 'higher-order'—tetrahedral 3D solid elements, which yield better results than linear elements, as stated in SOLIDWORKS-2020 HELP [30], because they represent curved boundaries more accurately and produce better mathematical approximations for the solid components. Finally, the whole 3D system was modelled by 249,470 tetrahedral finite elements. The number of finite elements (FE) for each part presented is as follows. The tyre has 162,156 FE, the rim has 62,367 the FE, the elastic inlet has 9979 FE and the basis has 14,968 FE.

## 3. Results and Discussion

The simulation can be divided into three stages. The first was the application of pressure to the tyre in a range from 0.18 MPa to 0.22 MPa. This stage finished after exactly one second. During this stage, contact between the tyre and the fixed basis occurred. This phenomenon was demonstrated clearly (see Figure 5) by increasing the normal force between the tyre and basis to the value ~1000 N. The second stage started after one second and continued until the ninth second. During this stage, the contact force was applied to the component responsible for the asphalt concrete basis, a vertical displacement towards the inflated tyre. The normal force $F_n$ started to increase from ~1000 N to ~6000 N, when the second stage finished. The tangential force component behaved slightly differently in the second stage (time range 1–9 s). It was flat from the beginning of the vertical movement of the fixed basis until the fifth second, increased by approximately a factor of three between the fifth and seventh seconds, and flattened again until the end of vertical movement at the ninth second. Finally, the starting tangential velocity component of basis movement began true tangential interaction between the tyre and the loaded basis. The quantitative character was slightly different for different external loads and internal tyre pressures (Figure 4, normal forces $F_{n18}$, $F_{n20}$, $F_{n22}$ and tangential forces $F_{t18}$, $F_{t20}$, $F_{t22}$, where the indices 18, 20 and 22 denote internal pressure inside a tyre).

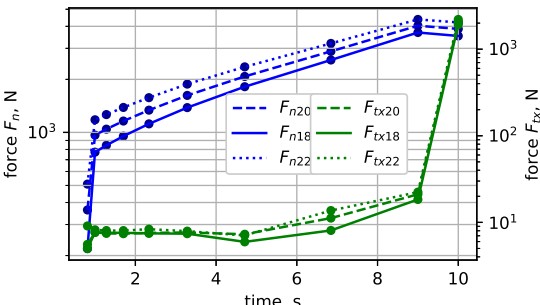

**Figure 5.** The model's resulting forces on tyre vs time for displacement of 15 mm, tyre internal pressure 0.18, 0.2 and 0.22 MPa and tyre–road friction coefficient 0.6.

Figure 6 presents velocities of the tread geometric centres in the x (a), z (b) and y (c) directions. Figure 7 shows the tread displacements which were used to calculate the tread velocities. Figure 6a shows that velocities $v_x$ of treads are distributed in opposite directions relative to the labelled eleventh tread (Figure 2a) in the centre of the tyre pattern. The velocity $v_x$ of the eleventh tread was slightly shifted to the x direction. Figure 6b demonstrates that the velocity $v_z$ was symmetrical about the pattern centre of the eleventh tread. Finally, Figure 6c shows the asymmetrical behaviour of the tread centre velocities in the y direction, which is normal to the road–tyre interface surface.

Next, friction coefficients were obtained for different tyre external load regimes, with different ratios of tangential and normal forces on the tread block. Six cases in total were investigated: three cases for ~4 kN external load and 0.2, 0.18 and 0.22 MPa internal tyre pressures (see Figure 8—the exact values of the normal external load were 3.869, 3.528 and 4.2 kN) and three cases for ~6 kN external load and 0.2, 0.18 and 0.22 MPa internal tyre pressures (see Figure 9—the exact values of the normal external load were 6.888, 6.517 and 7.283 kN). Each friction case in Figures 8 and 9 is followed by the normal and tangential force distribution over all of the road surface that made contact with the tread blocks. The 2D friction distribution shown in Figure 8 can be characterised by the mean and standard deviation values of 0.542448 ± 0.046138 for internal pressure 0.2 MPa, 0.556167 ± 0.038144 for internal pressure 0.18 MPa and 0.533904 ± 0.047833 for internal pressure 0.22 MPa, respectively. In the same manner, the 2D friction distribution in Figure 8 can be characterised by mean and standard deviation values of 0.472007 ± 0.085539 for internal pressure 0.2 MPa, 0.477212 ± 0.083172 for internal pressure 0.18 MPa and 0.449233 ± 0.089224 for internal pressure 0.22 MPa. A comparison of the results shows a 'flatter' distribution of friction coefficients of tread blocks

for external tyre load values equal to ∼4 kN, or approximately double the standard deviation of the overloaded tyre. Investigation of the normal and tangential force components on treads $F_n$, $F_t$ shows that the extreme values of normal force components on tread blocks 11 and 15 are at least 1.5 times bigger in the overloaded tyre compared with the non-overloaded tyre. A similar trend can be observed for the tangential force components.

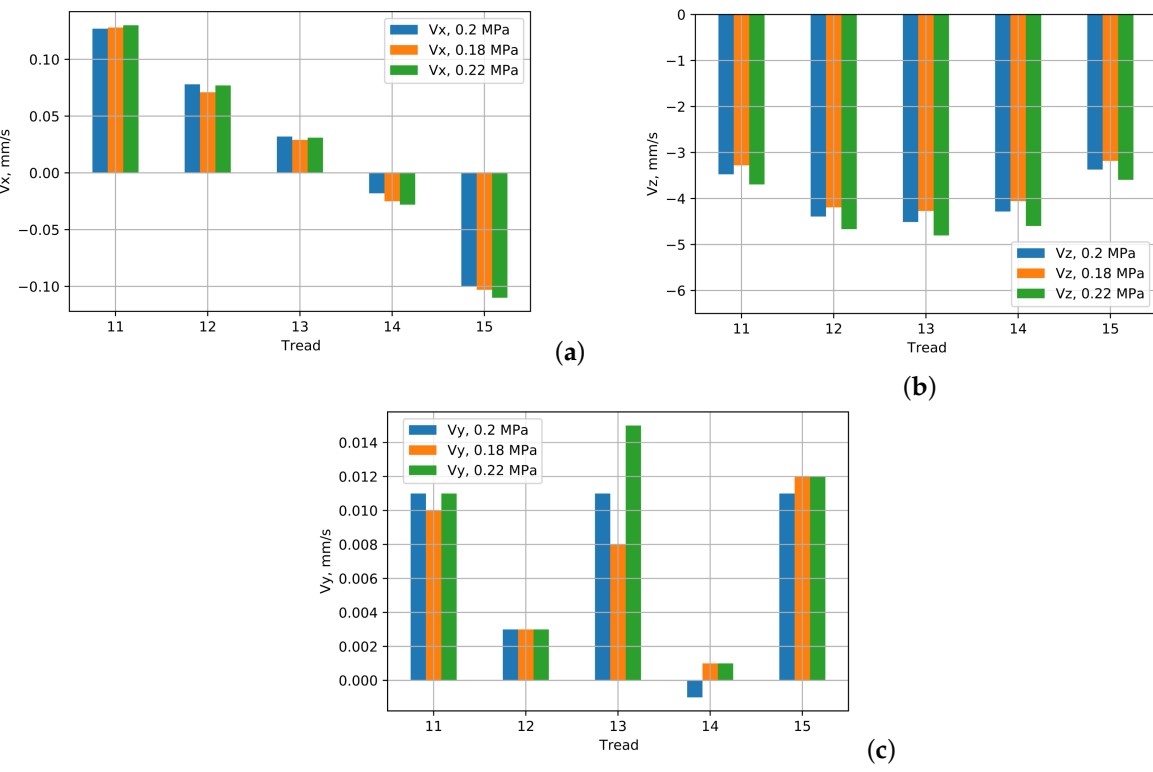

**Figure 6.** (**a**) Velocities of tread geometric centres in the x directions, (**b**) Velocities of tread geometric centres in the z directions, (**c**) Velocities of tread geometric centres in the y directions.

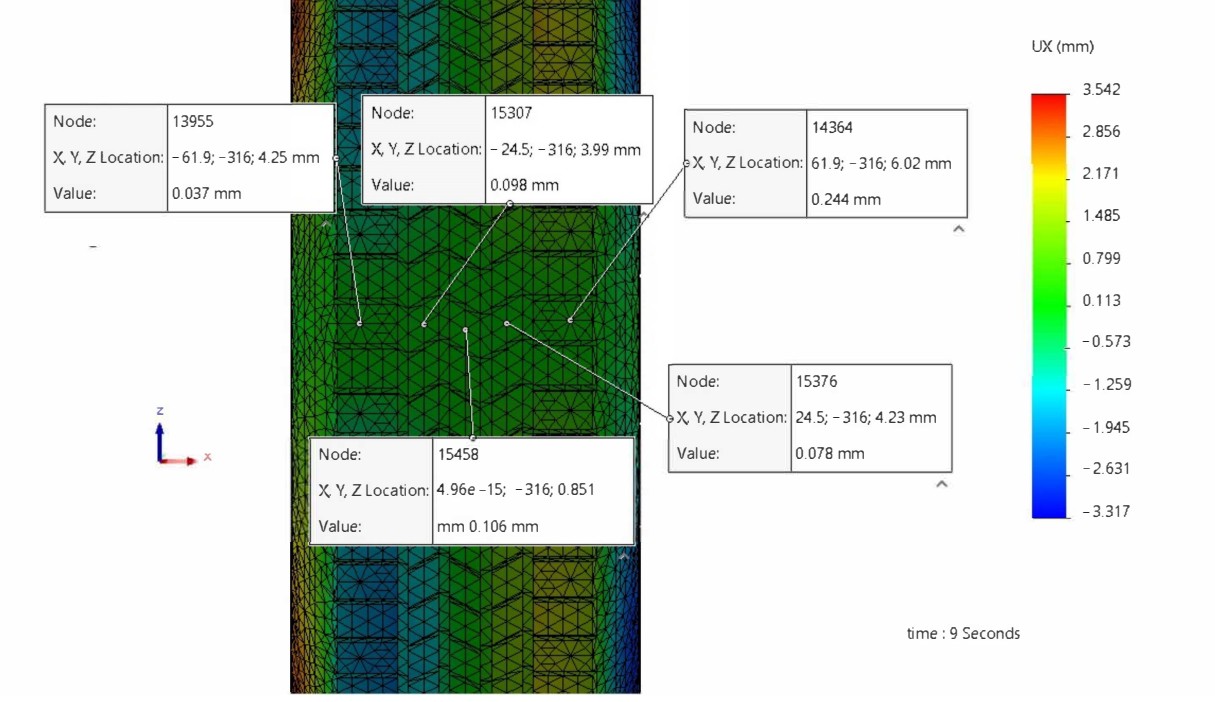

**Figure 7.** Displacements of tyre treads.

Finally, the friction $\mu_z$ distribution was calculated by projecting its 2D values of each tread block onto the x and z axes, where the x axis was perpendicular and the z axis was parallel to the tyre's translational moving direction, and the skewness of friction projection was obtained. The asymmetry index *AS* was calculated for left–right side normal and tangential transverse forces. All resulting values of skewness and asymmetry are presented in Table 2. Some tendencies can be observed by investigation of the obtained results. The skewness of friction projection onto the x coordinate was greater under the natural car's load onto a tyre. A consequence of this is the asymmetric pattern of the tyre tread block geometry. On the other hand, the z component of friction skewness was greater under the opposite conditions due to braking when the external load onto the front tyres increased to approximately 6 kN; however, this increase in skewness onto the z direction was smaller due to the rotational motion of the tyre. Furthermore, the modelling shows that the skewness is smallest with an internal pressure value of 0.18 MPa in both external load conditions. The same tendencies can be observed for normal and tangential force asymmetry.

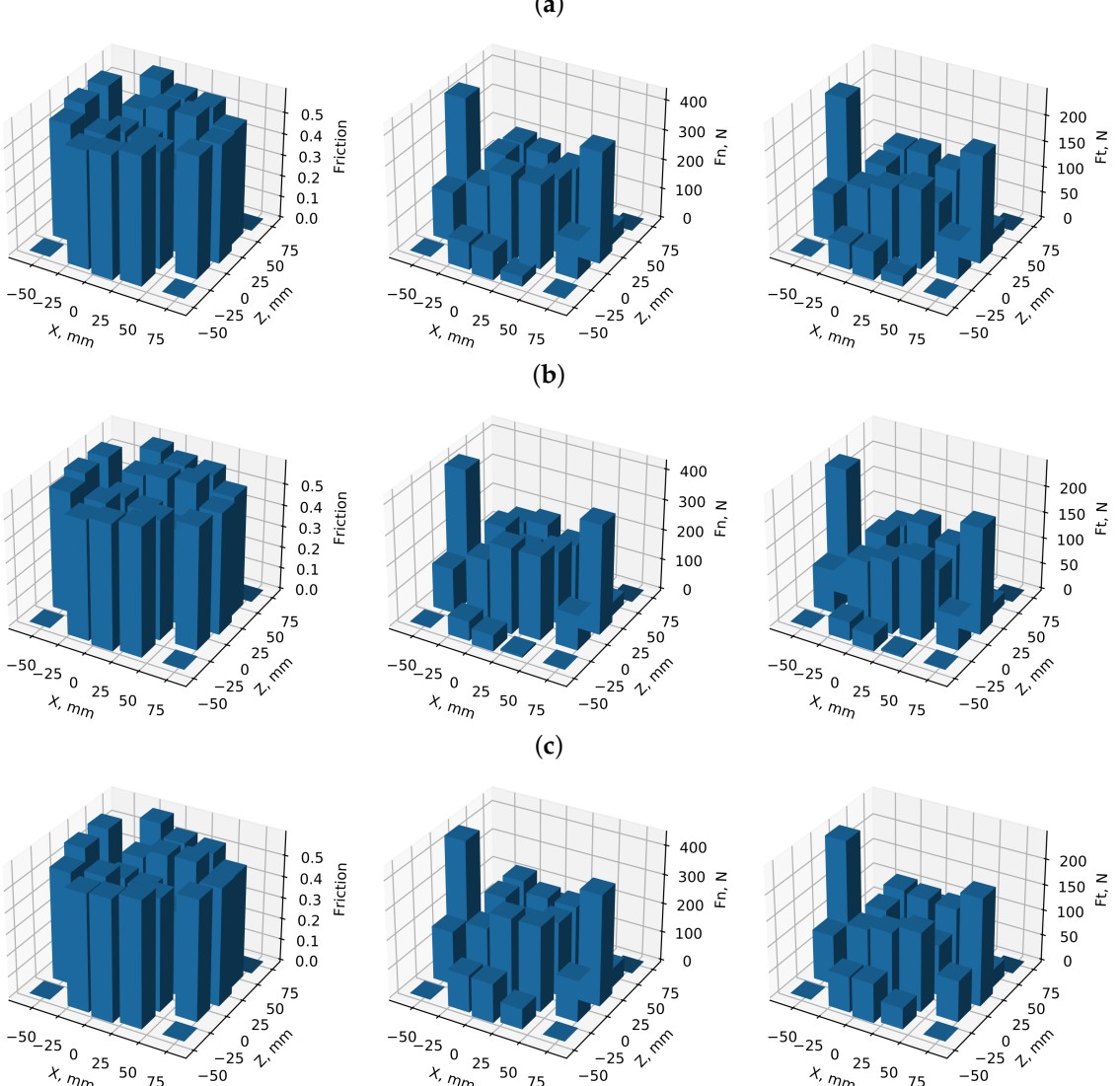

**Figure 8.** Friction $\mu_z$, normal $F_n$ and tangential $F_{tz}$ force distribution on the tyre–asphalt interface for pressures 0.2 (**a**), 0.18 (**b**) and 0.22 (**c**) MPa under a ∼4 kN load.

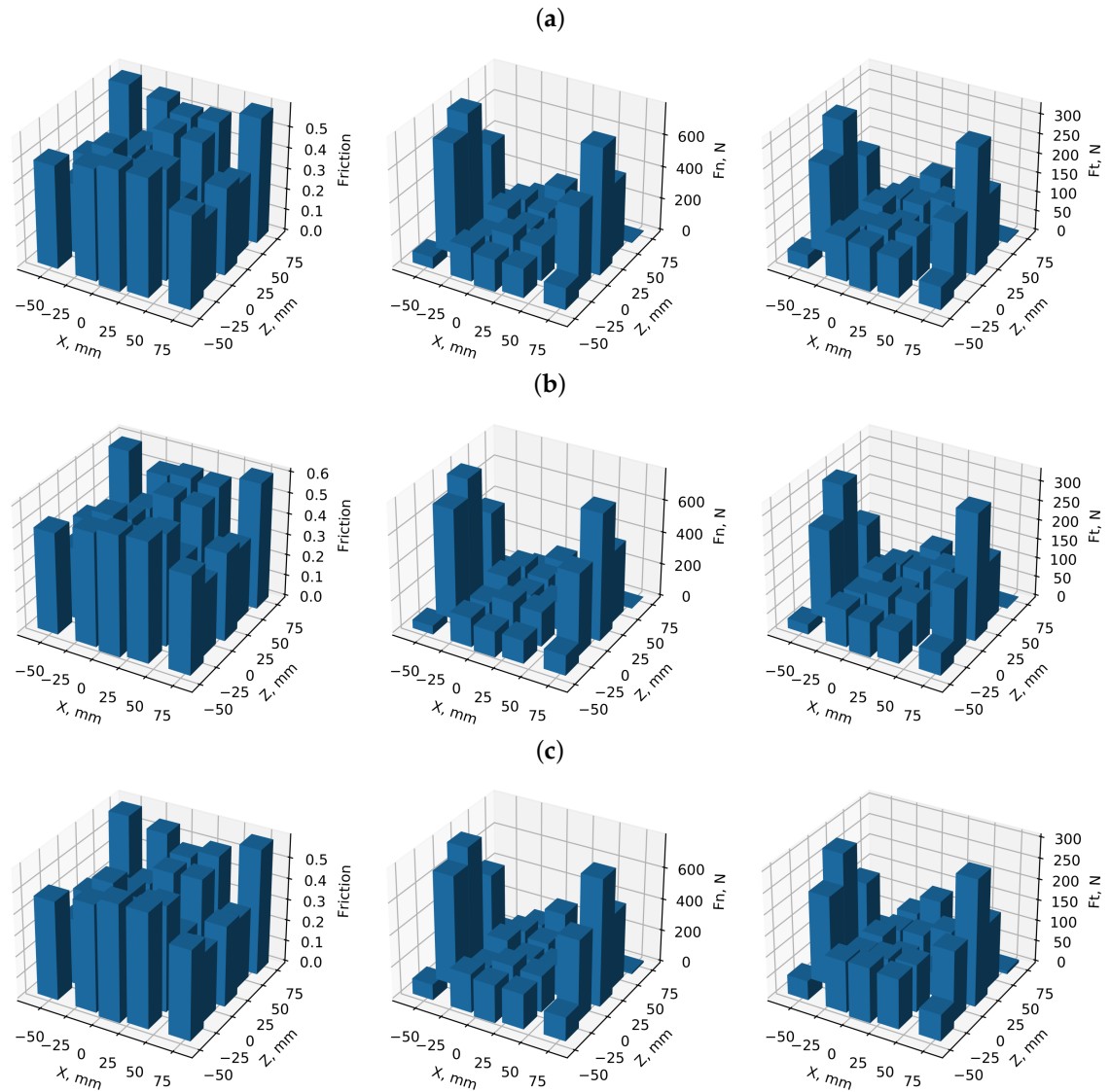

**Figure 9.** Friction $\mu_z$, normal $F_n$ and tangential $F_{tz}$ force distribution on the tyre–asphalt interface for pressures 0.2 (**a**), 0.18 (**b**) and 0.22 (**c**) MPa under a ∼6 kN load.

**Table 2.** Skewness of friction coefficients and normal force asymmetry.

| | Internal Pressure, MPa | | | | | | Load, kN |
|---|---|---|---|---|---|---|---|
| | 1.8 | | 2.0 | | 2.2 | | |
| Direction | x | z | x | z | x | z | |
| $\mu_z$ skewness | 0.0132 | 0.0179 | 0.0211 | 0.0127 | 0.0219 | 0.0098 | |
| $F_n$ asymmetry | 1.101 | - | 1.092 | - | 1.089 | - | 4 |
| $F_{tx}$ asymmetry | 0.979 | - | 0.985 | - | 1.015 | - | |
| $\mu_z$ skewness | 0.0004 | 0.0193 | 0.0144 | 0.0246 | 0.0148 | 0.0255 | |
| $F_n$ asymmetry | 1.070 | - | 1.068 | - | 1.064 | - | 6 |
| $F_{tx}$ asymmetry | 1.105 | - | 1.084 | - | 1.071 | - | |

## 4. Conclusions

A model of a car wheel was proposed which incorporates the detailed geometry of a rubber tyre with metallic reinforcement, a metallic rim and the road surface. The nonlinear dynamics of this model were analysed using the finite element method. The numerical

investigation of tyre deflection for normal and overloaded initial conditions led to the following conclusions.

1.  In the case of a slight asymmetry in the tread pattern of the tyre, in the contact area between the tyre and the road surface, the slip velocities of the treads at opposite points in the contact area (11 and 15) in the radial direction of the tyre were different in size and opposite signs. This suggests that the treads on different sides of the contact zone would generate different frictional forces and have different amounts of tread wear. Tread slip rates in the longitudinal contact zone of the tyre in the transverse section of the contact zone were almost the same. This trend of slip speeds was observed for different tyre pressures.

2.  In the case of a slight asymmetry in the tread pattern of the tyre, it was found that, in the area of contact between the tyre and the road surface, the vertical velocity distributions of the treads in the transverse direction of the tyre (11 to 15 treads) were almost the same.

3.  Due to the slight asymmetry of the tyre tread pattern, the contact normal and tangential tread forces at the edges of the tyre were higher than at the centre of contact of the tyre, and the right and left tread forces were different, a trend that persisted at different wheel vertical loads of 4 kN and 6 kN.

4.  Finally, when talking about the safety of a moving car, we should keep in mind the asymmetry of tyre tread patterns, which can influence instability due to asymmetric tangential forces in the tyre contact zone during braking when the tyre is in an overdeflected state.

**Author Contributions:** Conceptualisation, A.M., M.B. and R.K.; methodology, R.K.; software, O.A.; validation, O.A., A.M. and M.B.; formal analysis, A.M.; investigation, O.A.; resources, A.M.; data curation, M.B.; writing—original draft preparation, A.M.; writing—review and editing, M.B.; visualisation, A.M.; supervision, M.B.; project administration, R.K.; funding acquisition, R.K. All authors have read and agreed to the published version of the manuscript.

**Funding:** This research was funded by a Lithuanian Research Council grant (project registration Nr. MIP-17-233).

**Institutional Review Board Statement:** Not applicable.

**Informed Consent Statement:** Not applicable.

**Data Availability Statement:** Not applicable.

**Conflicts of Interest:** The authors declare no conflict of interest. The funders had no role in the design of the study; in the collection, analyses, or interpretation of data; in the writing of the manuscript; or in the decision to publish the results.

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
