# Peer review of "Modelling the Interaction between a Laterally Deflected Car Tyre and a Road Surface"

_applsci, doi:10.3390/app122211332_

Round 1

Reviewer 1 Report

Review:

The paper describes the Modelling of Deflected Car Tyre Interaction with Pavement. Suggestions and comments are as follows:

1.      The introduction is not sufficient. The aim of the paper is not precise and clear. The authors have made an overview of the other work, and there are no comparisons of how their work benefits from others. The introduction needs to be revised regarding the present aim and goal of the work.

2.      Why the interaction with pavement (sidewalk-British, hard surface as-America ENG. or type of the surface such as tarmac…) and not road uncertainty like different ground conditions, such as gaps, bumps, different ground types etc.? Clarify the title of the paper.

3.      What is the novelty of the model regarding the other?

4.      How does the tyre block’s shape and patterns influence the model?

Author Response

Answers to Reviewer I:

  1. 1.      The introduction is not sufficient. The aim of the paper is not precise and clear. The authors have made an overview of the other work, and there are no comparisons of how their work benefits from others. The introduction needs to be revised regarding the present aim and goal of the work.

The introduction is rewritten.

  1. 2.      Why the interaction with pavement (sidewalk-British, hard surface as-America ENG. or type of the surface such as tarmac…) and not road uncertainty like different ground conditions, such as gaps, bumps, different ground types etc.? Clarify the title of the paper.

The title of the paper is modified.

  1. What is the novelty of the model regarding the other?

The novelty of the paper is clarified. It reflects in improved introduction and in conclusion (the calculations of antisymmetric tyre thread, gradual loading of entire wheel model and the verification of centrifugal force in order to determine the skewness coefficient and force asymmetry index).

  1. How does the tyre block’s shape and patterns influence the model?

The tyre block’s shape was modelled in good agreement with real tyre. The asymmetric thread of the tyre influences the distribution of contact forces in case of sudden braking. We’ve added the extended explanation of obtained results in discussion and conclusions.

Reviewer 2 Report

1. Describe the organization of article at end of Introduction section

2. What are the parameters to consider for result section for investigate the influence of Tyre deflection on friction coefficients of the Tyre threads. 

3. Where does this work encounter problems, and what are those drawbacks? In what possible way can the comprehensiveness of this work be highlighted?

4. Author must describe the mathematical model for the Skewness of friction coefficients?

5. It would be helpful to have a deeper understanding of the findings. When attempting to demonstrate an investigation's real purpose, preliminary research at the research project level is required?

6. The paper's writing requires a great deal of effort in semantic parsing, spelling, and presentation. Because there are many typos and badly edited sentence fragments, the article takes proper English fine tuning.

Author Response

Answers to Reviewer II:

  1. Describe the organization of article at end of Introduction section

The organization of the article in introduction section was added.

  1. What are the parameters to consider for result section for investigate the influence of Tyre deflection on friction coefficients of the Tyre threads.

We’ve added the extended explanations on varied parameters in modelling section, such as tyre pressure, load value of external force.

  1. Where does this work encounter problems, and what are those drawbacks? In what possible way can the comprehensiveness of this work be highlighted?

The novelty of the paper is clarified. It reflects in improved introduction and in conclusion (the calculations of antisymmetric tyre thread, gradual loading of entire wheel model and the verification of centrifugal force in order to determine the skewness coefficient and force asymmetry index).

  1. Author must describe the mathematical model for the Skewness of friction coefficients?

Explanation added.

  1. It would be helpful to have a deeper understanding of the findings. When attempting to demonstrate an investigation's real purpose, preliminary research at the research project level is required?

This work proposes a methodology which would be useful for verification of contact forces in case of sudden braking. The extended explanation is added to the discussion.

  1. The paper's writing requires a great deal of effort in semantic parsing, spelling, and presentation. Because there are many typos and badly edited sentence fragments, the article takes proper English fine tuning.

English language was improved. The proofreading certificate of English article proofread is added.

Round 2

Reviewer 1 Report

Regarding the modification of the paper, the paper can be accepted in its present form. 

Reviewer 2 Report

All comments response properly.